# Effect of Carboxymethylcellulose Hyaluronan (SEPRAFİLM^®^) on an Arthrofibrosis Model Created in Rabbit Knees

**DOI:** 10.3390/life15091405

**Published:** 2025-09-05

**Authors:** Ismail Tugay Yagci, Ovunc Akdemir, Atilla Eyuboglu, Murat Sezak, Semih Aydogdu

**Affiliations:** 1Department of Orthopedics and Traumatology, Liv Hospital, Istanbul 34145, Turkey; tugay.yagci@icloud.com; 2Department Plastic, Aesthetic and Reconstructive Surgery, Istanbul Aydin University, Istanbul 34295, Turkey; 3Department Plastic, Aesthetic and Reconstructive Surgery, Istanbul Arel University, Istanbul 34010, Turkey; 4Department of Pathology, Ege University, Izmir 35100, Turkey; 5Department of Orthopedics and Traumatology, Ege University, Izmir 35100, Turkey

**Keywords:** adhesion reduction, arthrofibrosis, fibrosis prevention rabbit knee model, Seprafilm^®^

## Abstract

**Purpose:** This study aimed to evaluate the efficacy of carboxymethylcellulose (Seprafilm^®^) for the prevention and treatment of arthrofibrosis in rabbit knees, as well as to investigate its underlying mechanisms of action against fibrosis and adhesion formation. **Methods:** Sixteen male New Zealand white rabbits were randomly divided into two groups: a control group and a treatment group that received Seprafilm^®^ following surgically induced arthrofibrosis in the knee joint. Macroscopic and histological assessments were performed to evaluate adhesion, fibrosis, inflammation, and edema. **Results:** In the control group, macroscopic adhesion was severe in five rabbits (62.5%), moderate in two (25%), and minimal in one (12.5%). No macroscopic adhesion was observed in the Seprafilm^®^ group. The mean adhesion score was 2.5 ± 0.75 in the control group versus 0 in the treatment group (*p* < 0.001). Histologically, five rabbits (62.5%) in the control group showed significant fibrosis, and three (37.5%) showed moderate fibrosis, whereas all rabbits in the Seprafilm^®^ group exhibited only minimal fibrosis (*p* < 0.001). **Conclusions:** Seprafilm^®^ was effective in reducing both macroscopic and histological signs of adhesion and fibrosis in a rabbit arthrofibrosis model. These findings suggest its potential as a preventive and therapeutic agent in managing arthrofibrosis.

## 1. Introduction

The term *arthrofibrosis*, first coined by J. Vernon Luck, refers to a loss of joint range of motion caused by intra-articular scar tissue formation. The incidence of arthrofibrosis ranges from 2% to 35% [1,2], and its frequency is increasing, particularly among older patients and those with degenerative arthritis [3]. This condition disrupts normal knee kinematics and may lead to progressive degenerative changes [4].

Contributing factors include surgical technique, **the timing of surgery**, prolonged immobilization, concomitant ligament injuries, infection, and reflex sympathetic dystrophy [5]. Arthrofibrosis may arise not only following surgical interventions but also after nonsurgical trauma. It has been described in multiple joints including the knee, shoulder, ankle, elbow, and hand [6,7,8].

Pathophysiologically, arthrofibrosis results from an exaggerated fibrotic response during tissue repair, marked by excessive deposition of extracellular matrix that limits joint mobility [9]. Although histologically similar to other fibroproliferative disorders [10], arthrofibrosis is distinct in its joint-specific manifestation.

The pathological process begins with synovial inflammation and progresses to subsynovial fibrosis, capsular thickening, and eventual contracture [11]. This cascade is believed to be driven by immune cell activation following tissue injury, which triggers oxidative stress and the release of profibrotic cytokines such as transforming growth factor-β (TGF-β), platelet-derived growth factor (PDGF), and interleukin-1 (IL-1) [12,13].

Current treatment strategies—such as physical therapy, arthroscopic debridement, and manipulation under anesthesia—are often insufficient, particularly in preventing recurrence. Various pharmacological and biomaterial-based agents, including hydrogel microspheres, temperature-sensitive poloxamers, hyaluronic acid, carboxymethylcellulose, decorin, chitosan, rapamycin, and hydroxycamptothecin, have demonstrated anti-adhesive effects in experimental models [14,15,16,17,18,19,20,21]. However, none have been commercialized or adopted in clinical practice for joint surgery.

Hyaluronic acid alone is rapidly cleared from the application site, limiting its efficacy. To enhance retention and barrier function, it is chemically conjugated with carboxymethylcellulose to form Seprafilm^®^ (Genzyme Corp., Cambridge, MA, USA), a bioresorbable membrane widely used in abdominal and gynecological surgeries. Burns et al. reported that Seprafilm^®^ converts to a gel within 24–48 h and remains at the site for up to 7 days, acting as a physical barrier that prevents adhesions. The material is eventually degraded in the liver and excreted in urine within 28 days.

Although Seprafilm^®^ is effective in abdominal applications, its efficacy in preventing joint adhesions remains inconclusive. In a study by Hayashi et al. [17], Seprafilm^®^ reduced adhesion formation in a rabbit joint model. However, their analysis focused primarily on biomechanical and fibroblastic responses and lacked comprehensive histopathological evaluation.

Therefore, the present study aimed to evaluate the efficacy of Seprafilm^®^ in preventing and reducing arthrofibrosis in a rabbit knee model. Furthermore, we sought to elucidate its effects on histological markers of fibrosis, inflammation, and adhesion formation.

## 2. Materials and Methods

All experimental procedures were approved by the IACUC and carried out in compliance with the National Research Council’s guidelines for the humane care and use of laboratory animals at the Experimental Animal Production and Research Laboratory. Efforts were made to minimize animal suffering and to reduce the number of animals used in the study.

### 2.1. Sample Size and Animal Allocation

The sample size of 16 rabbits (8 per group) was determined based on previous similar studies (typically 6–10 animals per group). No formal statistical power analysis was performed, as this was an exploratory pilot study. Sixteen skeletally mature male New Zealand White rabbits (3500–4000 g, ≥6 months) were randomly assigned into two equal groups (n = 8). Male rabbits were chosen to avoid hormonal variability, which could influence fibrosis and adhesion outcomes.

Randomization was performed using a computer-generated sequence (Randomizer^®^, Austria) by an independent researcher to minimize allocation bias. Animals were housed in numbered cages without any other selection criteria.

**Group 1 (control) (n = 8):** No additional procedures were performed on the rabbit knee joint after the arthrofibrosis procedure.

**Group 2 (treatment) (n = 8):** After inducing arthrofibrosis in the rabbit knee joint similar to that in the control group, Seprafilm^®^ was placed in the knee joint.

### 2.2. Animal Husbandry and Environmental Control

All procedures were conducted in an IACUC-approved facility under controlled temperature (20–22 °C), humidity (50–60%), and a 12-h light/dark cycle. Rabbits were housed individually (5.43 ft^2^ floor area) and provided ad libitum access to water and a standardized high-fiber rabbit diet (Invigo 2031). Environmental enrichment, including chew toys and hiding spaces, was provided to reduce stress-related variables.

### 2.3. Standardization and Bias Control

To reduce interindividual variability, surgical and postoperative procedures were standardized and performed by a single surgeon. Animal health and behavior were monitored daily, with deviations recorded and assessed.

Histopathological evaluations were performed by a single pathologist blinded to the treatment groups to prevent assessment bias. While the operating surgeon was aware of group allocation, all subsequent evaluations were conducted under blinding protocols to minimize subjective influence.

This study was reported in accordance with the **ARRIVE guidelines** for in vivo experimental research.

### 2.4. Surgical Methods

A standardized surgical procedure was employed for both groups. Anesthesia was induced via intramuscular injection of a mixture of ketamine (Alfamine^®^ 10% injectable—ALFASAN) at 35 mg/kg and xylazine hydrochloride (Xylazol^®^—PROVET) at 8 mg/kg. To ensure consistency, the same surgeon performed all surgeries. The rabbits were placed in a supine position, and a longitudinal skin incision was made on the anterior aspect of the right knee, which was flexed at 90°. The quadriceps muscle was exposed, and the knee joint was accessed using a medial parapatellar approach with a parallel incision following the muscle fibers. The lateral and medial femoral condyles were exposed. Abrasion was performed using a No. 11 blade on the patellar joint surface and the anterior region of the supracondylar femur [17].

In Group 1, after the surgical procedure described above, capsulorrhaphy was performed with Vicryl 3-0, followed by skin closure with Nylon 4-0 sutures [22].

In Group 2, a sterile Seprafilm^®^ patch measuring 2.5 × 1.5 cm, intended to prevent adhesion formation, was placed in the supracondylar anterior femur region where abrasion was performed. Subsequently, capsulorrhaphy was performed with Vicryl 3-0, and skin closure was completed with Nylon 4-0 sutures. Following closure, the immobilization of the knee joint of rabbits in both groups was achieved with No. 5 wire sutures.

Six weeks after the procedure, all rabbits were euthanized with a high dose of intravenous barbiturate. Wire sutures were removed. For macroscopic and histopathological evaluation, soft tissues were dissected up to the medial condyle of the femur without damaging the extensor mechanism using a No. 11 blade (Figure 1). During the 6-week period following trauma induction, the animals were monitored daily. No mortality was observed among the rabbits’ post procedure.

### 2.5. Macroscopic Evaluation

A modified adhesion scoring system (MAS) was used to macroscopically assess adhesions in the suprapatellar region between the quadriceps and the anterior cortex of the femur [23]. This system categorizes adhesions into four grades based on their severity: Grade 0 indicates no adhesion present; Grade 1 denotes a weak, soft film-like adhesion that can be easily eliminated with minimal manual traction; Grade 2 represents a moderate adhesion that can be removed with manual traction; and Grade 3 signifies intense adhesion requiring surgical intervention for removal (Table 1). Range of motion was evaluated by passive flexion and extension of the knee joint under anesthesia. A goniometer was used for consistency; however, results were recorded qualitatively as presence or absence of restriction rather than in degrees.

### 2.6. Histopathological Evaluation

After fixation in 10% neutral buffered formalin for approximately 24 h, all connective tissues containing fibrotic adhesive scars were preserved, and the knee joints were decalcified in 20% formic acid for 48 h at room temperature. Four 5 mm-thick sections were obtained from each sample, starting 2 cm proximally to the femoral joint surface and extending into the suprapatellar pouch. Following routine tissue processing with a Leica ASP 3005, the samples were embedded in paraffin blocks. Eight 4 µm sections were cut from the paraffin blocks using a Leica RM 2145 microtome. Hematoxylin–eosin and Masson’s Trichrome staining was applied to the sections. Histological grading was performed according to previously published criteria [15,17,21]. Two blinded pathologists independently scored all samples, and interobserver agreement was >90%. Any discrepancies were resolved by consensus.

A single pathologist performed the evaluation in a blind manner. We employed a semiquantitative scoring system to assess the parameters. The presence of adhesions in rabbit knees, along with the intensity of fibrosis (fibroblast proliferation), inflammatory cell types (lymphocytes, neutrophils (PMNL), macrophages), edema, vascular proliferation, the giant cell response, and the formation and intensity of synovial chondrometaplasia, were semiquantitatively scored. These parameters (neutrophils, lymphocytes, macrophages, fibrosis, edema, vascular proliferation, giant cell response, and synovial chondrometaplasia) were semiquantitatively scored in 10 randomly selected fields at ×20 and ×100 magnification by a pathologist. The criteria for each score level were rigorously defined and applied uniformly.

#### 2.6.1. Distinguishing and Scoring of Cell Types

Neutrophils: Identified by their characteristic multilobed nuclei and pale-staining cytoplasm in hematoxylin-eosin (H and E)-stained sections.

Lymphocytes: Recognizable by their small, round nuclei and scant cytoplasm. The H and E sections generally exhibited a darker staining pattern.

Macrophages: These cells were identified by their larger, more irregular nuclei and abundant cytoplasm, which may also contain phagocytosed material.

The cells of each type were counted and scored semiquantitatively in 10 fields at ×20 and ×100 magnification. The scoring system ranged from 0 to 3, where 0 = absent, 1 = minimal, 2 = moderate, and 3 = abundant.

#### 2.6.2. Fibrosis

Fibroblast Proliferation: Evaluated based on the density of fibrous tissue and the extent of fibroblast proliferation. Areas with a high density of collagen and fibroblasts were scored as 3, while areas with minimal fibrosis were scored as 1. The scoring was performed semiquantitatively in the same 10 fields.

Identification: Fibrosis is characterized by the accumulation of dense fibrous connective tissue, predominantly collagen fibers. In H and E-stained sections, fibrosis appears as thickening or scarring of the tissue, often accompanied by an increase in fibroblast proliferation.

Evaluation technique: Areas with extensive collagen deposition and fibroblast proliferation were identified by their eosinophilic staining and dense appearance compared to those of the surrounding tissue. The degree of fibrosis was evaluated semiquantitatively, considering the density and distribution of the fibrous tissue.

#### 2.6.3. Edema

Edema was assessed based on the amount of interstitial fluid and tissue swelling observed. The severity of the condition was scored as 0 = none, 1 = mild, 2 = moderate, or 3 = severe.

Identification: Edema is characterized by an accumulation of interstitial fluid leading to swelling of the tissue. It can be observed as an increase in the space between cells and clear spaces or vacuoles can be observed in the tissue sections.

Evaluation technique: Edema is identified by the presence of these spaces or vacuoles in the tissue. The severity of edema was scored based on the extent of swelling observed in the tissue sections.

#### 2.6.4. Vascular Proliferation

The extent of vascularization was evaluated by counting the number of newly formed blood vessels. Vascular proliferation was scored as 0 = absent, 1 = few, 2 = moderate, or 3 = numerous.

Identification: vascular proliferation involves the formation of new blood vessels within the tissue. This is typically observed as an increased number of capillaries or small blood vessels in the tissue.

Evaluation technique: vascular proliferation was assessed by counting the number of new blood vessels in the tissue sections. The presence of newly formed vessels was noted, and the extent of vascularization was scored based on the number of vessels and their distribution.

#### 2.6.5. Giant Cell Response

The presence of multinucleated giant cells was noted, and the extent of their formation was scored similarly to other parameters: 0 = absent, 1 = few, 2 = moderate, and 3 = many.

Identification: Giant cells are multinucleated cells formed by the fusion of macrophages in response to chronic inflammation. They are identified by their large size and multiple nuclei.

Evaluation technique: The presence of giant cells was noted by their distinctive morphology. The extent of giant cell formation was assessed semiquantitatively based on the number and size of the giant cells present in the tissue.

#### 2.6.6. Formation and Intensity of Synovial Chondrometaplasia

Synovial Chondrometaplasia: The formation and intensity of this morphology were assessed based on the presence and distribution of cartilaginous metaplasia within the synovial tissue. The scores ranged from 0 = none to 1 = focal, 2 = moderate, and 3 = extensive.

Identification: Synovial chondrometaplasia involves the formation of cartilage-like tissue within the synovial membrane. It is identified by the presence of cartilage nodules or islands within the synovial lining.

Evaluation technique: The formation and intensity of chondrometaplasia were assessed based on the number and size of these cartilage formations within the synovial membrane. The presence and extent of chondrometaplasia were scored according to the distribution and intensity of the cartilage-like tissue.

Quality Control: To ensure accuracy and consistency in histopathological scoring, the pathologist was trained and calibrated in the scoring system before the evaluations. Interobserver variability was minimized by standardizing the evaluation procedures. Regular quality checks were incorporated to maintain adherence to the defined scoring criteria.

## 3. Statistical Analysis

All statistical analyses were performed using SPSS version 16.0 (IBM Corp., Armonk, NY, USA). Data were expressed as median and interquartile range (IQR) unless otherwise indicated. The Mann–Whitney U test was employed to compare histopathological parameters—namely fibrosis (fibroblast proliferation), vascular proliferation, synovial chondrometaplasia, giant cell response, edema, and inflammatory cell infiltration (lymphocytes; neutrophils; macrophages)—as well as macroscopic adhesion scores between the Seprafilm^®^ and control groups. A *p*-value < 0.05 was considered statistically significant.

## 4. Results

### 4.1. Macroscopic Adhesion Scores

Adhesions were assessed using the Modified Adhesion Scoring (MAS) system. In the control group, 62.5% of rabbits exhibited severe adhesions (score: 3), 25% moderate (score: 2), and 12.5% minimal (score: 1). No adhesions were observed in the Seprafilm^®^ group (score: 0 in all animals). The median adhesion score was significantly higher in the control group (median: 2.5, IQR: 2–3) compared to the Seprafilm^®^ group (median: 0, IQR: 0–0; *p* < 0.001; Table 2, Figure 2). All rabbits in the treatment group demonstrated full range of motion, whereas all animals in the control group exhibited restricted mobility of the knee joint.

### 4.2. Microscopic Adhesion

Microscopic adhesions were detected in all rabbits in the control group, while none were observed in the Seprafilm^®^ group (*p* < 0.0001), corroborating the macroscopic findings.

### 4.3. Fibrosis (Fibroblast Proliferation)

Severe fibroblast proliferation was observed in 62.5% of control rabbits and moderate in 37.5%, whereas all Seprafilm^®^-treated rabbits exhibited only minimal fibrosis. Median fibrosis scores were significantly higher in the control group (median: 2, IQR: 2–3) compared to the Seprafilm^®^ group (median: 1, IQR: 1–1; *p* < 0.001).

### 4.4. Vascular Proliferation

In the control group, 25% of rabbits showed moderate vascular proliferation and 37.5% minimal, while the Seprafilm^®^ group exhibited 50% minimal and 50% absent vascular proliferation. The median vascular proliferation score was significantly higher in the control group (median: 2, IQR: 1–2) than in the Seprafilm^®^ group (median: 1, IQR: 0–1; *p* < 0.001).

### 4.5. Synovial Chondrometaplasia

Synovial chondrometaplasia was present in 62.5% of the control group (25% moderate, 37.5% minimal) and in 37.5% of the Seprafilm^®^ group (all minimal). Although chondrometaplasia was more frequent in the control group, the difference was not statistically significant (control: median: 1, IQR: 0–2; Seprafilm^®^: median: 0, IQR: 0–1; *p* > 0.05).

### 4.6. Edema

Edema was observed in 50% of control animals (12.5% severe, 12.5% moderate, 25% minimal), while 62.5% of Seprafilm^®^ animals exhibited no edema and 37.5% had minimal edema. The control group had significantly higher edema scores (median: 1, IQR: 0–2) than the Seprafilm^®^ group (median: 0, IQR: 0–1; *p* < 0.001).

### 4.7. Neutrophil (PMNL) Infiltration

Neutrophilic infiltration was abundant (score 3) in 87.5% and moderate (score 2) in 12.5% of control group rabbits. No neutrophilic infiltration was observed in the Seprafilm^®^ group. Median scores were significantly higher in the control group (median: 3, IQR: 3–3) compared to the Seprafilm^®^ group (median: 0, IQR: 0–0; *p* < 0.001).

### 4.8. Lymphocyte Infiltration

Lymphocyte infiltration showed no significant difference between groups. In the control group, 50% showed minimal, 37.5% moderate, and 12.5% severe infiltration. In the Seprafilm^®^ group, 75% had minimal and 25% moderate infiltration (control: median: 1, IQR: 1–2; Seprafilm^®^: median: 1, IQR: 1–1; *p* > 0.05).

### 4.9. Macrophage Infiltration

Macrophage infiltration also did not differ significantly between groups. In the control group, 37.5% of rabbits had minimal, 50% moderate, and 12.5% severe infiltration. In the Seprafilm^®^ group, infiltration was minimal in 37.5%, moderate in 12.5%, severe in 12.5%, and absent in 37.5% (control: median: 2, IQR: 1–2; Seprafilm^®^: median: 1, IQR: 0–2; *p* > 0.05). (Table 3)

## 5. Discussion

Arthrofibrosis (AF) is characterized by excessive periarticular fibrosis and symptomatic restriction of joint motion due to an exaggerated immune response following proinflammatory insults. The hallmark of AF is capsular contracture resulting from abnormal extracellular matrix (ECM) deposition, which may occur secondary to trauma, surgery, infection, or hemarthrosis. Although rare, idiopathic AF may also arise in the absence of a clear trigger [24,25,26]. For instance, Ryan Will et al. identified arthrofibrosis of the proximal interphalangeal joint as a significant postoperative complication of trigger finger release surgery [27].

While the pathogenesis of AF remains incompletely understood, studies suggest a potential genetic predisposition, especially in patients with Dupuytren’s disease [10]. Fibroproliferative disorders such as Dupuytren’s, Ledderhose, and Peyronie’s disease involve heightened TGF-β expression and aberrant collagen synthesis [28,29,30]. Given the histological overlap with these conditions, our results may offer preliminary insights into the intra-articular application of anti-adhesive barriers such as Seprafilm^®^ in managing broader fibrotic pathologies.

In previous studies, intermittent hyaluronic acid injection or TGF-β neutralizing antibody infusion was reported to reduce adhesion formation [18,31]. However, these approaches carry a risk of postoperative infection due to repeated administration. Seprafilm^®^, a bioresorbable membrane composed of hyaluronic acid and carboxymethylcellulose, offers a safer alternative as a single perioperative application [17]. Initially developed to prevent intra-abdominal adhesions, Seprafilm^®^ has demonstrated efficacy in gynecologic, colorectal, and pelvic surgeries [32,33,34].

In an arthrofibrosis model similar to ours, Hayashi et al. reported improved knee extension following Seprafilm^®^ application but did not explore histological changes [17]. Our study, in contrast, provides a comprehensive assessment including both macroscopic and histological analyses. In the Seprafilm^®^-treated group, we observed significantly reduced adhesion and fibrosis, alongside lower edema and neutrophil infiltration levels. However, lymphocyte and macrophage infiltration, vascular proliferation, and synovial chondrometaplasia showed no significant differences, likely reflecting the chronic phase of inflammation at the six-week evaluation point. The antifibrotic effect of Seprafilm^®^ can be attributed both to its physical role as a mechanical barrier preventing tissue adhesion and to the biological properties of hyaluronic acid. HA is known to modulate inflammatory responses, reduce fibroblast proliferation, and limit extracellular matrix deposition, which may contribute to the reduced fibrosis observed in our study.

A strong correlation was identified between macroscopic adhesion, microscopic adhesion, and fibroblast proliferation, suggesting that fibrosis severity directly contributes to adhesion formation. Additionally, neutrophil infiltration correlated moderately with lymphocyte, macrophage, and edema scores, indicating interconnected inflammatory pathways.

Our findings align with prior studies. For example, Brunelli et al. demonstrated a significant reduction in adhesions using Hyaloglide^®^, a hyaluronic acid gel lacking carboxymethylcellulose [22]. In our study, Seprafilm^®^ completely prevented adhesion formation. Similarly, Fukui et al. showed that high-dose decorin reduced fibrous adhesions, though without significant histological differences [18]. Ariyan et al. described histological similarities between Dupuytren’s disease, Peyronie’s disease, and capsular contracture of the breast—notably increased fibroblasts, edema, and neutrophils—paralleling our control group findings [35].

Masson’s Trichrome staining revealed marked differences between groups. Control specimens exhibited dense collagen deposition, disorganized muscle fibers, fibrovascular proliferation, and inflammation. In contrast, the Seprafilm^®^ group showed preserved muscle architecture, minimal interstitial fibrosis, and fewer neovessels, indicating both anti-fibrotic and tissue-preserving effects. Reduced chondrometaplastic changes were also observed in this group (Figure 3).

The observed outcomes suggest that Seprafilm^®^ may confer mechanical and biological benefits by limiting fibroblast activation, dampening inflammation, and preserving extracellular matrix structure. Further molecular studies are needed to evaluate its role in cytokine modulation and tissue remodeling.

The sample size of eight rabbits per group was deemed sufficient based on previous studies using similar arthrofibrosis models and was considered adequate to demonstrate significant differences in adhesion and fibrosis outcomes [36]. This supports the robustness of our findings while adhering to ethical guidelines.

Limitations include the small sample size and species-specific responses in rabbits, which may limit generalizability. Moreover, the six-week follow-up may not fully reflect long-term outcomes. Functional metrics and immune profiling were also beyond this study’s scope. Nevertheless, the study adhered to ARRIVE guidelines, strengthening its reliability [37].

Our results highlight Seprafilm^®^’s potential as a prophylactic agent in preventing arthrofibrosis after intra-articular surgery. Complete inhibition of adhesion in the treatment group suggests promising translational value, particularly for procedures like total knee arthroplasty. Future research should investigate long-term effects, molecular pathways, and clinical applicability in humans. Future studies could include midterm sacrification and histopathological evaluation (at 2–3 weeks), while Seprafilm^®^ is still present in the joint cavity, to better elucidate the temporal dynamics of its anti-fibrotic effects.

## 6. Conclusions

This study demonstrated that carboxymethylcellulose-based Seprafilm^®^ significantly reduces both macroscopic and microscopic adhesions and fibrosis in a rabbit arthrofibrosis model. The membrane’s anti-inflammatory and anti-fibrotic properties suggest it may be a valuable adjunct in preventing joint complications following surgery. Despite limitations, the findings justify further exploration in larger animal models and early-phase clinical trials. Future work should aim to clarify its long-term safety, dose-responsiveness, and potential for modulating intra-articular inflammation and fibrosis in human applications.

## Figures and Tables

**Figure 1 life-15-01405-f001:**
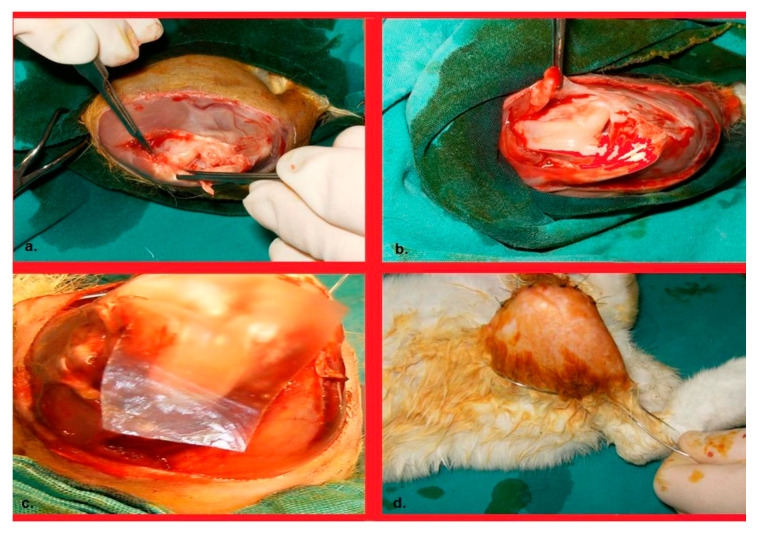
(**a**). Talasization of the patellar joint surface and the supratubercular area of the anterior cortex of the femur. (**b**). Her appearance after completion of the talasization procedures; and (**c**). Image of the Seprafilm placement. (**d**). Image of a wire suture passing through the groin and ankle posteriorly, providing immobilization of the knee in flexion.

**Figure 2 life-15-01405-f002:**
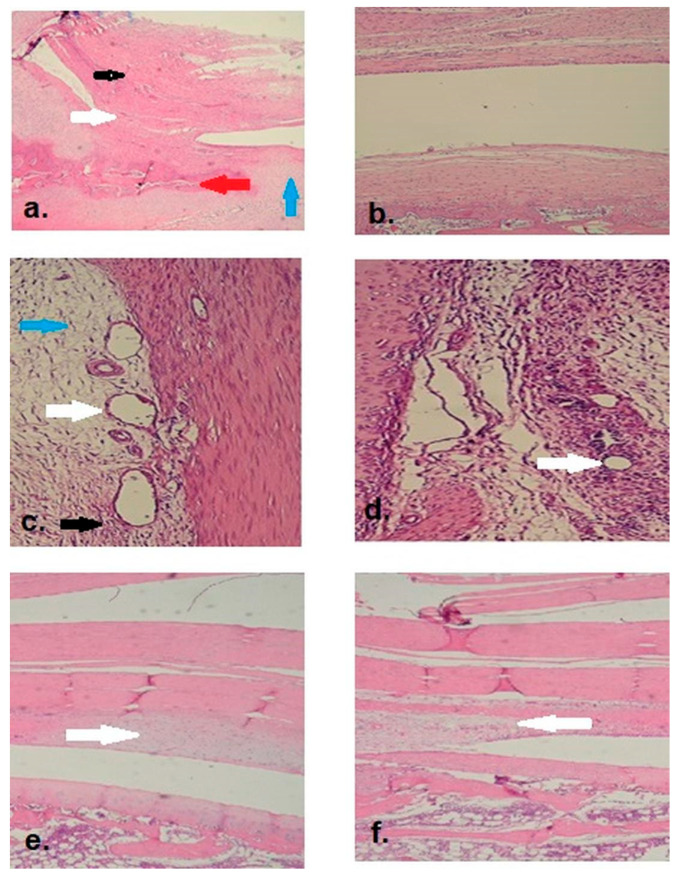
(**a**). Adhesion observed in the suprapatellar region of the control group (2× magnification). (Black arrow: quadriceps tendon, white arrow: adhesive tissue and fibrosis, blue arrow: cartilage tissue, red arrow: bone tissue). (**b**). No adhesion or fibrosis was observed in the suprapatellar space of the Seprafilm^®^ group (2× magnification). (**c**). Adhesive tissue with fibrovascular features was observed in the control group, along with pronounced edema, vascular dilation and proliferation, fibroblasts, and predominant neutrophil (PMNL) inflammatory cells (20× magnification). (White arrows: vascular dilation and proliferation; black arrow: neutrophil (PMNL) infiltration; blue arrow: edema). (**d**). Minimal vascular dilation and proliferation were observed in the Seprafilm^®^ group (20× magnification). (White arrows: vascular dilation and proliferation). (**e**). Appearance of moderate synovial chondrometaplasia below the quadriceps tendon in the control group (2× magnification). (White arrow: synovial chondrometaplasia area). (**f**). Appearance of minimal synovial chondrometaplasia below the quadriceps tendon in the Seprafilm^®^ group (2× magnification). (White arrow: synovial chondrometaplasia area).

**Figure 3 life-15-01405-f003:**
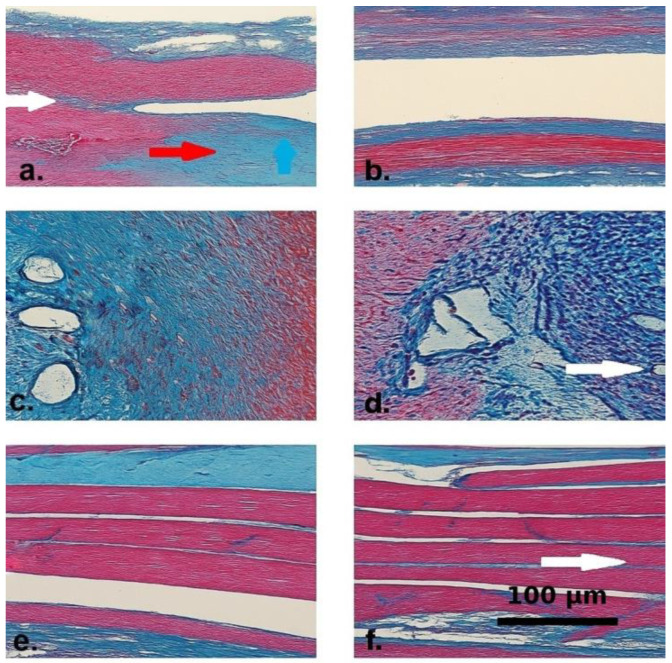
Masson’s Trichrome-stained histological sections from the control (**a**,**c**,**e**) and Seprafilm^®^-treated (**b**,**d**,**f**) groups. Collagen fibers are stained blue, muscle fibers appear red, and nuclei are darkly stained. (**a**) Control group specimen showing interstitial collagen accumulation (blue arrow) and dense fibrotic bands within the extracellular matrix (red arrow). The white arrow marks a disrupted transition zone between muscle fibers and surrounding fibrotic tissue, indicating architectural disorganization. (**b**) Seprafilm^®^-treated tissue demonstrating preserved muscle fiber alignment and minimal interstitial collagen deposition. (**c**) Control section displaying prominent fibrovascular proliferation and dense inflammatory infiltration around dilated vascular structures. (**d**) Seprafilm^®^-treated sample with visibly reduced inflammatory infiltration and neovascularization. The white arrow indicates the absence of vascular dilation and highlights regular vascular morphology. (**e**) Control group tissue showing misaligned muscle bundles separated by dense intermuscular fibrotic septa. (**f**) Seprafilm^®^-treated section with parallel and intact muscle fibers. The white arrow denotes preserved intermuscular architecture with minimal fibrotic septation and nearly normal histological appearance.

**Table 1 life-15-01405-t001:** The macroscopic evaluation based on the modified adhesion scoring (MAS) system used in the study.

Score	Description
**0**	No adhesion—absence of adhesion between the quadriceps and the anterior cortex of the femur.
**1**	Weak, soft film-like adhesion—adhesion present but easily eliminated with minimal manual traction.
**2**	Moderate adhesion—adhesion present and can be eliminated with manual traction.
**3**	Intense adhesion—adhesion is intense and requires surgical intervention for removal.

**Table 2 life-15-01405-t002:** The results of the macroscopic adhesion scoring in the control group rabbits compared to the Seprafilm^®^ group.

Group	MAS: 0 (No Adhesion)	MAS: 1 (Minimal)	MAS: 2 (Moderate)	MAS: 3 (High Level)	Mean MAS	Mann–Whitney U *p*-Value
**Control Group**	N/A	1 rabbit (12.5%)	2 rabbits (25%)	5 rabbits (62.5%)	2.5 ± 0.75	*p* < 0.0001
**Seprafilm^®^ Group**	All rabbits (100%)	N/A	N/A	N/A	0	N/A

“MAS” refers to the macroscopic adhesion score. The percentages represent the proportion of rabbits in each group with the corresponding MAS. “Mean MAS” represents the average macroscopic adhesion score in each group. “Mann–Whitney U *p*-value” indicates the statistical significance of the difference in adhesion presence between the control and Seprafilm^®^ groups.

**Table 3 life-15-01405-t003:** The histopathological findings and statistical analysis results between the control group and the Seprafilm^®^ group for various parameters.

Parameter	Control Group	Seprafilm^®^ Group	Median (IQR) Control/Seprafilm^®^	*p*-Value
**Microscopic Adhesion**	Detected in all rabbits	Not detected in any rabbits	–/–	<0.0001
**Fibrosis** (Fibroblast Proliferation)	High (62.5%), Moderate (37.5%)	Minimal (100%)	2 (2–3)/1 (1–1)	<0.001
**Vascular Proliferation**	Moderate (25%), Minimal (37.5%), None (37.5%)	Minimal (50%), None (50%)	2 (1–2)/1 (0–1)	<0.001
**Synovial Chondrometaplasia**	Moderate (25%), Minimal (37.5%), None (37.5%)	Minimal (37.5%), None (62.5%)	1 (0–2)/0 (0–1)	>0.05
**Edema**	Significant (12.5%), Moderate (12.5%), Minimal (25%), None (50%)	None (62.5%), Minimal (37.5%)	1 (0–2)/0 (0–1)	<0.001
**Neutrophil Infiltration (PMNL**	|Significant (87.5%), Moderate (12.5%)	None (100%)	3 (3–3)/0 (0–0)	<0.001
**Lymphocyte Infiltration**	Minimal (50%), Moderate (37.5%), Significant (12.5%)	Minimal (75%), Moderate (25%), Significant (0%)	1 (1–2)/1 (1–1)	>0.05
**Macrophage Infiltration**	Minimal (37.5%), Moderate (50%), Significant (12.5%)	Minimal (37.5%), Moderate (12.5%), Significant (12.5%), None (37.5%)	2 (1–2)/1 (0–2)	>0.05

“*p*-values” represent the level of statistical significance in comparisons between the control and Seprafilm^®^ groups for each histopathological parameter. A *p*-value < 0.05 was considered statistically significant. PMNL, polymorphonuclear leukocyte.

## Data Availability

The data generated and/or analyzed during the current study are stored by the corresponding author on an external hard drive and are also available in the Ege University institutional database. The datasets can be obtained from the corresponding author or from Ege University upon reasonable request. Additionally, the datasets are available in the Ege University repository at: https://acikerisim.ege.edu.tr (accessed on 18 April 2018).

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
