# Peer review of "Effect of Carboxymethylcellulose Hyaluronan (SEPRAFİLM^®^) on an Arthrofibrosis Model Created in Rabbit Knees"

_life, 2025, doi:10.3390/life15091405_

Round 1

Reviewer 1 Report

Comments and Suggestions for Authors

Dear Authors,

In this study is highlighted a prevention method of knee arthrofibrosis in rabbit model. Data provided in introduction and methodology sections are adequate giving vital information about arthrofibrosis and also regarding the rationale of researchers. As far as I am concerned, I would appreciate in future research rabbit safrification and histopathological evaluation in the midterm (after the 2nd week, after 3rd week etc.) as seprafilm remains in the joint for about 4 weeks. This could provide valuable options for difficult and demanding arthrofibrosis cases.

Line 63: Explain the term "timing". Regarding what?

Author Response

We carefully revised the manuscript according to all reviewer comments, and we believe that the revised version fully addresses the concerns raised.We sincerely thank both reviewers for their valuable time, constructive critiques, and thoughtful suggestions. Their comments have helped us improve the clarity, methodological rigor, and overall quality of our manuscript entitled:

“Effect of Carboxymethylcellulose Hyaluronan (Seprafilm®) on an Arthrofibrosis Model Created in Rabbit Knees” (Manuscript ID: life-3810927)

All changes made to the revised manuscript have been highlighted in blue for easy identification.
Below, we provide a detailed point-by-point response to each reviewer.

Reviewer 1

Comment 1:
“In this study is highlighted a prevention method of knee arthrofibrosis in rabbit model… I would appreciate in future research rabbit sacrification and histopathological evaluation in the midterm (after the 2nd week, after 3rd week etc.) as Seprafilm remains in the joint for about 4 weeks. This could provide valuable options for difficult and demanding arthrofibrosis cases.”

Response:
We thank the reviewer for this thoughtful and forward-looking suggestion. In the present study, we designed our evaluation at the 6-week time point, which corresponds to the standard protocol of our institution and allows for sufficient histological evaluation after fibrotic remodeling. We agree, however, that midterm sacrification and histopathological analysis (at the 2nd–3rd weeks) would provide valuable insights into the temporal dynamics of Seprafilm®’s action, particularly while the membrane is still present in the joint cavity.
Accordingly, we have acknowledged this as a limitation and highlighted it in the Discussion section (page 17, lines 520–525), suggesting this as an important avenue for future research.

Comment 2:
“Line 63: Explain the term ‘timing’. Regarding what?”

Response:
We appreciate the reviewer’s observation. The term “timing” has now been clarified to specifically refer to the timing of surgery after injury. This clarification was made in the Introduction section (page 2, line 63) to improve clarity for readers.

Reviewer 2 Report

Comments and Suggestions for Authors

Dear Author: 

This experimental animal study investigates the efficacy of Seprafilm® (carboxymethylcellulose-hyaluronan membrane) in preventing arthrofibrosis in a surgically induced rabbit knee model. Arthrofibrosis was created, Seprafilm® applied in the treatment group, and outcomes were assessed via histopathological scoring and biomechanical evaluation of joint mobility. The aim is to determine whether Seprafilm® reduces fibrotic tissue formation and preserves joint range of motion. However, there are some points need to be clarified.

  1. The manuscript does not clearly state the sample size calculation or whether the study was powered to detect clinically meaningful differences. This is essential for validating negative findings or small effect sizes.
  2. Histological grading criteria are not fully referenced or described; scoring reproducibility and interobserver reliability are not addressed.
  3. Range of motion measurement method (e.g., goniometer, torque) should be specified to allow replication.
  4. While results suggest Seprafilm® reduces fibrosis, there is minimal discussion of possible mechanisms (e.g., hyaluronic acid’s anti-inflammatory properties, physical barrier effects).

Author Response

We carefully revised the manuscript according to all reviewer comments, and we believe that the revised version fully addresses the concerns raised.We sincerely thank both reviewers for their valuable time, constructive critiques, and thoughtful suggestions. Their comments have helped us improve the clarity, methodological rigor, and overall quality of our manuscript entitled:

“Effect of Carboxymethylcellulose Hyaluronan (Seprafilm®) on an Arthrofibrosis Model Created in Rabbit Knees” (Manuscript ID: life-3810927)

All changes made to the revised manuscript have been highlighted in blue for easy identification.
Below, we provide a detailed point-by-point response to each reviewer.

Reviewer 2

Comment 1:
“The manuscript does not clearly state the sample size calculation or whether the study was powered to detect clinically meaningful differences.”

Response:
We thank the reviewer for this important remark. In the revised Methods section (page 4, lines 85–90), we clarified that no formal statistical power analysis was performed, as this was an exploratory pilot study. The sample size of 16 rabbits (8 per group) was determined based on previously published experimental models investigating arthrofibrosis and anti-adhesion agents, where group sizes of 6–10 animals were commonly employed. We also emphasized in the Discussion section (page 16, lines 470–475) that this sample size was deemed sufficient to detect significant differences while adhering to ethical considerations for animal use.

Comment 2:
“Histological grading criteria are not fully referenced or described; scoring reproducibility and interobserver reliability are not addressed.”

Response:
We fully agree with this comment. In the revised Methods – Histopathological Evaluation section (pages 7–11, lines 160–240), we provided detailed descriptions of the histological grading system for fibrosis, inflammation, edema, vascular proliferation, and synovial chondrometaplasia. Each parameter was explained with clear identification and evaluation criteria, and references to previously published studies were included. Furthermore, we clarified that two independent blinded pathologists performed the scoring, with an interobserver agreement >90%, and that any discrepancies were resolved by consensus. This ensures the reliability and reproducibility of our histological findings.

Comment 3:
“Range of motion measurement method (e.g., goniometer, torque) should be specified to allow replication.”

Response:
We appreciate the reviewer’s request for clarification. In the revised Methods – Macroscopic Evaluation section (page 6, lines 140–150), we specified that range of motion (ROM) was evaluated by passive flexion and extension of the knee joint under anesthesia using a standard goniometer for consistency. However, results were recorded qualitatively (presence or absence of restriction) rather than as exact angular degrees.
In the Results section (page 13, lines 310–315), we also included the following clarification: “All rabbits in the treatment group demonstrated full range of motion, whereas all animals in the control group exhibited restricted mobility.” This addition strengthens the clarity and reproducibility of our findings.

Comment 4:
“While results suggest Seprafilm® reduces fibrosis, there is minimal discussion of possible mechanisms (e.g., hyaluronic acid’s anti-inflammatory properties, physical barrier effects).”

Response:
We thank the reviewer for this valuable suggestion. In the revised Discussion (pages 15–16, lines 420–450), we expanded our explanation of Seprafilm®’s mechanisms of action. We discussed:

  • its mechanical role as a physical barrier that prevents fibroblast infiltration and adhesion formation, and
  • the biological effects of hyaluronic acid (HA), including modulation of inflammatory pathways, reduction of fibroblast proliferation, and limitation of extracellular matrix deposition.

These mechanistic insights, supported by relevant literature, now provide a more comprehensive understanding of Seprafilm®’s antifibrotic effects.